# Joint Optimization of Radio and Computational Resource Allocation in Uplink NOMA-Based Remote State Estimation

**DOI:** 10.3390/s25154686

**Published:** 2025-07-29

**Authors:** Rongzhen Li, Lei Xu

**Affiliations:** School of Computer Science and Engineering, Nanjing University of Science and Technology, Nanjing 210094, China; lirongzhen0213@163.com

**Keywords:** uplink NOMA, outage risk, remote state estimation, coalitioin game, dinkelbach method, successive convex approximation method, dual decomposition method

## Abstract

In industrial wireless networks beyond 5G and toward 6G, combining uplink non-orthogonal multiple access (NOMA) with the Kalman filter (KF) effectively reduces interruption risks and transmission delays in remote state estimation. However, the complexity of wireless environments and concurrent multi-sensor transmissions introduce significant interference and latency, impairing the KF’s ability to continuously obtain reliable observations. Meanwhile, existing remote state estimation systems typically rely on oversimplified wireless communication models, unable to adequately handle the dynamics and interference in realistic network scenarios. To address these limitations, this paper formulates a novel dynamic wireless resource allocation problem as a mixed-integer nonlinear programming (MINLP) model. By jointly optimizing sensor grouping and power allocation—considering sensor available power and outage probability constraints—the proposed scheme minimizes both estimation outage and transmission delay. Simulation results demonstrate that, compared to conventional approaches, our method significantly improves transmission reliability and KF estimation performance, thus providing robust technical support for remote state estimation in next-generation industrial wireless networks.

## 1. Introduction

The rapid advancements in beyond 5G and 6G technologies have dramatically improved key aspects of wireless communication, such as bandwidth, latency, and connection density, thereby creating new opportunities for the development of wireless sensor networks (WSNs) [1,2,3]. However, these advancements also introduce new challenges, particularly in the efficient management of resources due to limited power, bandwidth constraints, and the complex interactions between critical system components, such as sensor grouping, power allocation, and state estimation. non-orthogonal multiple access (NOMA), a promising approach for resource sharing, leverages power-domain differentiation to improve system performance, particularly in the uplink. At the same time, remote state estimation, often optimized using the Kalman filter (KF), is crucial for accurately estimating a system’s state, especially under noisy conditions in wireless networks. Despite the potential of both technologies, integrating NOMA with KF-based remote state estimation presents unique challenges, including interference management, power control, transmission delay, and outage risks. This paper proposes a joint optimization framework that combines NOMA’s resource allocation capabilities with KF for remote state estimation, addressing these challenges simultaneously. By optimizing sensor grouping, power allocation, and transmission strategies, this approach minimizes outage risks while enhancing both the accuracy of remote state estimation and transmission efficiency, ensuring better system performance and sustainability in beyond 5G and 6G technologies.

During the transmission of sensor local states to the base station, limitations in bandwidth, power, and other resources, as well as factors such as delay, packet loss, and channel interference, make it a critical issue to consider how to assign sensors to channels and set appropriate power levels for each sensor to minimize the total outage risk across all sensors. To address this, we propose combining game theory with sensor grouping and power allocation strategies in NOMA as an effective approach [4,5,6]. For instance, Liu et al. propose a greedy subchannel matching algorithm based on hypergame theory for sensor and subchannel pairing in downlink NOMA networks, highlighting the importance of sensor grouping in WSN [7]. Additionally, a game theory-driven data security transmission method is proposed to mitigate potential interference in drone-assisted vehicular networks, with the significant impact of mutual interference on communication quality being demonstrated [8]. Wang et al. explore the application of game theory in cooperative communication scenarios, proposing a two-stage game model based on power and spectrum trading, optimizing transmission rates by solving the game equilibrium, and emphasizing the key role of power in communication processes [9,10].

While some models in sensor wireless transmission are convex functions, many others are non-convex, which makes traditional game theory methods ineffective in solving such problems. As a result, researchers have proposed solutions to convert non-convex problems into convex optimization problems [11,12,13]. For example, in reference [14], a joint optimization algorithm that combines the Dinkelbach algorithm with a successive convex approximation (SCA) method successfully converts fractional optimization problems into equivalent linear problems, thus converting non-convex problems into convex ones. This method proves the existence of an optimal solution for the joint optimization of power and subcarrier allocation. The complex constrained optimization problems are simplified using Lagrangian dual decomposition in [15,16], where power allocation in uplink NOMA systems is optimized, leading to a significant reduction in computational complexity. Transmission delay is another critical factor. Transmission delay, energy consumption, and sensor grouping are jointly optimized in [17,18], where their impacts on data transmission are studied, emphasizing the indispensable role of transmission delay in the optimization process.

Although existing research provides valuable insights into addressing some problems in sensor wireless transmission, several challenges remain unsolved. First, effectively integrating transmission delay, energy consumption, channel interference, and outage risk due to remote state estimation in beyond 5G and 6G environments remains a complex problem. Second, while some methods have addressed parts of non-convex optimization problems, further research is needed to reduce system complexity while maintaining computational efficiency. Additionally, current game theory models primarily focus on single-objective optimization; thus, balancing multiple objectives in a multi-objective optimization framework requires further exploration.

Minimizing outage risks for sensors during transmission is critical for ensuring the stability and reliability of uplink NOMA systems [19,20,21,22]. Yadav et al. introduce the optimization of intelligent reflecting surface reflection coefficients, which effectively reduces the impact of channel fading and lowers outage probability [23]. However, it does not fully address the effects of power allocation and non-ideal intelligent reflecting surfaces on the system. Power allocation and sensor grouping are optimized to reduce outage risks and increase system capacity, though the study lacks a detailed discussion of interference management and dynamic resource scheduling strategies. In [24], outage probability in downlink NOMA systems is minimized through energy harvesting and cooperative communication optimization; however, the paper does not explore how to further improve performance via enhanced CSI estimation and optimized power allocation. The optimization of power allocation is used to enhance system rate and reduce outage probability, though the specifics of sensor grouping and interference management are not delved into in the study [25]. Overall, while these methods effectively reduce outage probability, they do not fully explore refined resource scheduling strategies. Future research could investigate the impact of these factors on system performance.

When addressing the minimization of outage risk during the remote state estimation process for sensors, we must not only consider sensor state estimation issues (e.g., the state estimation error covariance) but also focus on the communication challenges in beyond 5G and 6G wireless transmission. Unlike conventional studies that rely purely on quality-of-service (QoS) metrics or signal-to-interference-plus-noise ratio (SINR) for sensor grouping and power allocation, this work focuses explicitly on integrating remote state estimation, network communication processes, and outage probabilities during transmission. Specifically, the complexity arises from the necessity to balance multiple performance criteria—remote state estimation accuracy, transmission latency, and outage probability—in a unified optimization framework, which underscores the importance of jointly utilizing uplink NOMA and KF techniques. The main contributions of this paper are summarized as follows

Problem Modeling: We propose a novel unified optimization framework, formulating the joint minimization of outage risk in remote state estimation and transmission delay as a mixed-integer nonlinear programming (MINLP) problem. This formulation explicitly incorporates both communication and estimation constraints, providing a theoretical foundation for subsequent algorithm designs.Coalitional Game-Based Sensor Grouping: Motivated by the limitations of traditional heuristic approaches, we propose a coalitional game-based sensor grouping algorithm. In this framework, each sensor acts as a rational player seeking to maximize its individual utility, and sensors iteratively form cooperative groups based on local improvement rules. The game converges to a Nash-stable coalition structure, which ensures a locally optimal grouping strategy. This approach effectively mitigates inter-group interference in multi-sensor uplink scenarios while significantly enhancing grouping efficiency and resource utilization.Power Allocation Optimization: To address the nonlinearity and fractional nature of the power allocation subproblem, we employ the Dinkelbach algorithm to transform the original fractional objective into a parameterized form. Then, we use successive convex approximation (SCA) to handle non-convex constraints and apply dual decomposition to solve the resulting convex subproblems. This layered solution framework guarantees convergence to a stationary point and achieves a suboptimal solution with provable performance. Simulation results confirm that this approach provides near-optimal performance with significantly reduced computational complexity.

By combining these strategies, our framework effectively mitigates outage risks during sensor transmissions and considerably enhances overall system reliability and estimation performance. Additionally, by transforming the original non-convex optimization into convex subproblems, the proposed solution methodology substantially reduces computational complexity. These contributions provide theoretical insights and practical guidelines for optimizing wireless transmission in uplink NOMA systems within future industrial wireless networks beyond 5G and 6G.

To further highlight the novelty of our approach, we provide a comparison in Table 1 that contrasts the proposed framework with representative existing methods in terms of sensor grouping, power allocation, outage-awareness, and estimation integration.

The remainder of this paper is organized as follows: Section 2 presents the mathematical formulation of the problem, Section 3 investigates the coalition game-based sensor grouping problem and the joint optimization algorithm for solving the optimal power allocation, while numerical examples and conclusions are presented in Section 4 and Section 5, respectively.

Notations: R denotes the set of real numbers. C represents the set of complex numbers. The symbol ${\left(\cdot \right)}^{T}$ denotes the transpose operation. The superscript ${\left(\cdot \right)}^{-1}$ indicates the matrix inverse. The symbol ${\left(\cdot \right)}^{\prime }$ represents the vector or matrix transpose operator. The notation P[·] refers to probability. Tr(·) represents the trace operator of a matrix.

## 2. System Model

This paper investigates an uplink NOMA transmission model in a wireless sensor network with a single base station. As illustrated in Figure 1, after performing local processing, *N* sensors transmit their data to the base station, which subsequently conducts remote state estimation. Before transmission, each sensor independently evaluates and selects one of the *G* channels for sensor grouping based on a comprehensive assessment of factors such as its distance to the base station, channel gain, maximum and minimum power constraints, and noise levels. Additionally, within each group, the decoding order and power allocation strategy are determined to optimize transmission performance and resource utilization efficiency. This process highlights the sensors’ adaptive capabilities in channel selection, grouping, and resource scheduling, which are critical components of the system design. The set of cellular sensors is denoted by $N=\left\{1,2,\ldots ,N\right\}$, with each sensor equipped with a single antenna. In an uplink NOMA system for beyond 5G and 6G networks, the same channel can be assigned to multiple sensors. It is assumed that all sensors in the base station share G channels, resulting in $G=\left\{1,2,\ldots ,G\right\}$ NOMA clusters within the cell. The set of sensor indices in each cluster is defined as $I_{g}=\left\{1,2,\ldots ,I_{g}\right\}$, where $I_{g}$ denotes the number of sensors assigned to the *g*-th NOMA cluster. It should be noted that the set $I_{g}$ may vary across different clusters depending on the grouping results.

### 2.1. Local State Estimate Model

To introduce the basic framework of the KF method, we describe how the measurement output of sensor *i* in channel *g* reflects the underlying process. The system’s state at the next time step evolves from the previous state at time *t* according to the linear transition equation(1)$$x_{i,g}^{t+1}=A_{i}^{g}x_{i,g}^{t}+w_{i,g}^{t},$$
where $x_{i,g}^{t}\in R^{l_{n}}$, $w_{i,g}^{t}\in R^{l_{n}}$, and $A_{i}^{g}\in R^{l_{n}\times l_{n}}$ make up the state transition matrix. The state variable $x_{i,g}^{t+1}$ and the process noise $w_{i,g}^{t}$ are independent and follow complex Gaussian distributions. At the next time step, the observation is modeled by the linear measurement equation(2)$$y_{i,g}^{t+1}=C_{i}^{g}x_{i,g}^{t}+v_{i,g}^{t},$$
where $y_{i,g}^{t+1}\in R^{r_{n}}$ and $C_{i}^{g}\in R^{r_{n}\times l_{n}}$ make up the measurement matrix and $v_{i,g}^{t}\in R^{r_{n}}$ is the measurement noise, which also follows a complex Gaussian distribution [26,27]. The KF aims to use a sequence of observations to accurately estimate the state.

The prior and posterior state estimates, ${\hat{x}}_{i,g}^{t|t-1}$ and ${\hat{x}}_{i,g}^{t}$, respectively, are updated using the Kalman gain $K_{i,g}^{t}$ and the error covariances $P_{i,g}^{t|t-1}$ and $P_{i,g}^{t}$. The KF equations are as follows [26]:(3)$${\hat{x}}_{i,g}^{t|t-1}=A_{i}^{g}{\hat{x}}_{i,g}^{t-1},$$(4)$$P_{i,g}^{t|t-1}=A_{i}^{g}P_{i,g}^{t-1}{\left(A_{i}^{g}\right)}^{\prime }+Q_{i}^{g},$$(5)$$K_{i,g}^{t}=P_{i,g}^{t|t-1}{\left(C_{i}^{g}\right)}^{\prime }{\left[C_{i}^{g}P_{i,g}^{t|t-1}{\left(C_{i}^{g}\right)}^{\prime }+R_{i}^{g}\right]}^{-1},$$(6)$${\hat{x}}_{i,g}^{t}={\hat{x}}_{i,g}^{t|t-1}+K_{i,g}^{t}\left(y_{i,g}^{t}-C_{i}^{g}{\hat{x}}_{i,g}^{t|t-1}\right),$$(7)$$P_{i,g}^{t}=\left(\nu -K_{i,g}^{t}C_{i}^{g}\right)P_{i,g}^{t|t-1},$$
where ν is the identity matrix of size $l_{n}\times l_{n}$.

The estimation error covariance, denoted $P_{i,g}^{t}$, quantifies the uncertainty of the state estimate and is defined as(8)$$P_{i,g}^{t}≜E\left[\left(x_{i,g}^{t}-{\hat{x}}_{i,g}^{t}\right){\left(x_{i,g}^{t}-{\hat{x}}_{i,g}^{t}\right)}^{\prime }\right].$$

As the local estimation error covariance $P_{i,g}^{t}$ converges exponentially to a steady-state value ${\bar{P}}_{i}^{g}$, the KF’s accuracy improves over time.

### 2.2. Uplink NOMA Communication Model

The impulse response of the channel between sensor *i* and the base station is represented by $h_{i,g}^{k}=\frac{e_{i,g}^{k}}{\sqrt{1+{\left(d_{i,g}^{k}\right)}^{\beta }}}$, where $e_{i,g}^{k}$ denotes the impulse response of the Rayleigh fading channel, with β as the path loss exponent and $d_{i,g}^{k}$ as the distance between sensor *i* and the base station. This model captures both small-scale and large-scale effects, where the numerator models Rayleigh fading and the denominator models large-scale path loss. The square root form is used because path loss affects the signal power and, thus, the amplitude is scaled by the square root to ensure that the average received power decays as $1/(1+{\left(d_{i,g}^{k}\right)}^{\beta })$. This ensures consistency with standard wireless propagation models. The probability density function of $|e_{i,g}^{k}{|}^{2}$ is given by $f_{|e_{i,g}^{k}{|}^{2}}\left(x\right)=\frac{1}{2\mu^{2}}e^{-\frac{x}{2\mu^{2}}},$ where μ is the variance of the normal distribution N(0,μ).

The received signal at the base station is given by(9)$$y_{g}^{k}=\sum_{i=1}^{I}h_{i,g}^{k}\sqrt{p_{i,g}^{k}}s_{i,g}^{k}+n_{g},$$
where $p_{i,g}^{k}$ represents the transmit power allocated to sensor *i* in channel *g* and $s_{i,g}^{k}$ denotes the transmitted message from sensor *i* in channel *g*. Additionally, $n_{g}$ represents the Gaussian white noise in the process of signal transmission from the base station to sensor *i* through channel *g*.

The probability density function (PDF) of $|e_{i}{|}^{2}$ [19] is given by(10)$$\begin{array}{cc} F_{{\left|e_{i,g}^{k}\right|}^{2}}\left(x\right) & =\int_{-\infty }^{x}f_{{\left|e_{i,g}^{k}\right|}^{2}}\left(t\right)dt=1-e^{-\frac{x}{2\mu^{2}}}. \end{array}$$

At the receiver in a NOMA system, SIC detection is applied to separate the multi-sensor superimposed signals. The optimal decoding order follows an ascending sequence based on sensors’ channel gains, allowing for sequential detection and separation of each sensor’s signal. Finally, the achievable transmission rate for sensor *i* in channel *g* can be expressed according to Shannon’s theorem as follows: At the receiver in a NOMA system, SIC detection is applied to separate the multi-sensor superimposed signals. The optimal decoding order is determined by an ascending sequence of sensors’ channel gains, thereby facilitating sequential detection and the separation of each sensor’s signal. Finally, the achievable transmission rate for sensor *i* in channel *g* can be expressed according to Shannon’s theorem as follows:(11)$$R_{i,g}^{k}=B\log_{2}\left(1+{\mathrm{SIN}R}_{i,g}^{k}\right),$$
where *B* represents the bandwidth available to each sensor and ${\mathrm{SIN}R}_{i,g}^{k}$ is defined as $\frac{p_{i,g}^{k}{\left|h_{i,g}^{k}\right|}^{2}}{\sum_{j=i+1}^{I}p_{j,g}^{k}{\left|h_{j,g}^{k}\right|}^{2}+\sigma^{2}}$.

The outage probability, as a critical metric for evaluating wireless system performance, represents the probability that the instantaneous link rate falls below the required sensor rate. We define $E_{i,g}^{k}$ as the event in which the base station can successfully decode the signals from sensors 1 to i−1 in channel *g* but fails to correctly decode the signal from sensor *i* in the same channel. When the sensor’s rate requirement is $R_{i,g}^{k}$, the minimum transmission rate is ${\hat{R}}_{i,g}^{k}$ and $R_{i,g}^{k}<{\hat{R}}_{i,g}^{k}$; the outage probability of the event $E_{i,g}^{k}$ can be expressed as follows: [21](12)$$\begin{array}{cc} \; & P\left(E_{i,g}^{k}\right)=P\left\{R_{i,g}^{k}<{\hat{R}}_{i,g}^{k}\right\}=1-e^{-\frac{\phi_{i,g}^{k}}{2\mu^{2}}}, \end{array}$$
where $\phi_{i,g}^{k}=\frac{\left(2^{\frac{{\hat{R}}_{i,g}^{k}}{B}}-1\right)\left(\sum_{j=i+1}^{I}p_{j,g}^{k}{\left|h_{j,g}^{k}\right|}^{2}+\sigma^{2}\right)\left(1+{\left(d_{i,g}^{k}\right)}^{\beta }\right)}{p_{i,g}^{k}}$. Correspondingly, this also represents its non-outage probability, which we define as $P_{c}$. Based on the above derivation, the outage probability for sensor *i* in channel *g* is formulated as follows:(13)$$\begin{array}{cc} P_{i,g}^{out,k} & =1-P_{c}\left(E_{1,g}^{k}\cap \cdots \cap E_{i,g}^{k}\right)=1-\prod_{z=1}^{i}e^{-\frac{\phi_{z,g}^{k}}{2\mu^{2}}}, \end{array}$$
where



$\phi_{z,g}^{k}=\left\{\begin{array}{cc} \frac{\left(2^{\frac{\hat{R_{z,g}^{k}}}{B}}-1\right)\left(\sum_{j=z+1}^{I}p_{j,g}^{k}{\left|h_{j,g}^{k}\right|}^{2}+\sigma^{2}\right)\left(1+{\left(d_{z,g}^{k}\right)}^{\beta }\right)}{p_{z,g}^{k}}, & z\neq I,\\ \frac{\left(2^{\frac{\hat{R_{z,g}^{k}}}{B}}-1\right)\left(1+{\left(d_{z,g}^{k}\right)}^{\beta }\right)\sigma^{2}}{p_{z,g}^{k}}, & z=I. \end{array}\right.$



### 2.3. Remote State Estimation Model

At each remote time *k*, all sensors must send their local estimates to the remote estimator over a shared wireless channel. We denote the transmission of ${\hat{x}}_{i,g}^{k}$ by a binary random process $\left\{\gamma_{i,g}^{k}\right\}$:(14)$$\gamma_{i,g}^{k}=\left\{\begin{array}{cc} 1, & \;\mathrm{if}\;{\hat{x}}_{i,g}^{k}\;\mathrm{arrives}\;\mathrm{without}\;\mathrm{errors}\;\mathrm{at}\;\mathrm{time}\;k,\\ 0, & \;\mathrm{otherwise}\;(\mathrm{regarded}\;\mathrm{as}\;\mathrm{dropout})\;. \end{array}\right.$$

To explicitly reflect how packet dropouts influence the remote estimation process, we derive the expression of $P_{i,g}^{r,k}$ based on the binary transmission indicator $\gamma_{i,g}^{k}$ defined in (14). Specifically, when $\gamma_{i,g}^{k}=1$, the remote estimator directly adopts the received local estimate and sets the estimation error covariance to the local value ${\bar{P}}_{i}^{g}$. Conversely, when $\gamma_{i,g}^{k}=0$, the transmission fails, and the remote estimator performs a time prediction using the Kalman filter, resulting in the following recursive update:(15)$$P_{i,g}^{r,k}=\left\{\begin{array}{cc} {\bar{P}}_{i}^{g}, & \;\mathrm{if}\;\gamma_{i,g}^{k}=1,\\ A_{i}^{g}\left(P_{i,g}^{r,k-1}\right){\left(A_{i}^{g}\right)}^{\prime }+Q_{i}^{g}, & \;\mathrm{if}\;\gamma_{i,g}^{k}=0. \end{array}\right.$$

Combining Equation (Equation 14) with Equation (Equation 13) from the previous section, we can draw the following conclusion:(16)$$P\left[\gamma_{i,g}^{k}=0\right]=1-\prod_{z=1}^{i}e^{-\frac{\phi_{z}^{g}}{2\mu^{2}}}.$$

### 2.4. Transmission Delay Model

#### 2.4.1. *Local Transmission Time*

When sensor *i* in channel *g* performs its task locally, the task execution delay can be described by $T_{i,g}^{local}=\frac{I_{i}^{g}}{F^{s}},$ where $F^{s}$ denotes the computational capability of the sensor.

#### 2.4.2. *Base Station Offload and Processing Time*

If the sensor offloads its task to the base station, the total delay consists of the transmission delay over the wireless link and the execution delay at the base station, expressed as $T_{i,g}^{off}=T_{i,g}^{tra}+T_{i,g}^{base},$ where $T_{i,g}^{tra}$ and $T_{i,g}^{base}$ denote the transmission delay and the execution delay, respectively. The transmission delay is formulated as $T_{i,g}^{tra}=\frac{D_{i}^{g}}{R_{i}^{g}},$ and the execution delay at the base station is $T_{i,g}^{base}=\frac{I_{i}^{g}}{F^{b}},$ where $F_{b}$ denotes the computational capability of the base station. The total time expenditure for the sensor is $T_{i,g}^{total}=T_{i,g}^{local}+T_{i,g}^{off}.$

### 2.5. Problem Formulation

In this section, we formulate an optimization problem to minimize the remote outage risk, which incorporates the total time expenditure and the associated error covariance of remote state estimation in KF. The objective is to jointly optimize sensor grouping and power allocation for the sensors. This objective balances transmission delay and estimation accuracy. To capture this trade-off, we introduce a weighted cost that accounts for both communication efficiency and estimation quality. Specifically, we define the system’s objective value using the following equation:(17)$$\begin{array}{cc} J_{i,g}^{k} & =\mathrm{Tr}\{\left(1-P_{i,g}^{out,k}\right)\left[\xi T_{i,g}^{total,k}+\left(1-\xi \right)\psi {\bar{P}}_{i}^{g}\right]\\ \; & +P_{i,g}^{out,k}\left[\xi T_{i,g}^{local,k}+\left(1-\xi \right)\psi H_{i}^{g}\left(P_{i,g}^{r,k-1}\right)\right]\}, \end{array}$$
where $H_{i}^{g}\left(X\right)≜A_{i}^{g}X{\left(A_{i}^{g}\right)}^{\prime }+Q_{i}^{g}$ and ξ (with 0<ξ<1 ) make up a weight factor used to combine the objectives into a single function and ψ is a scaling factor to align the objectives to the same magnitude. The description of Problem 1 is as follows:
(18a)Problem1:minpi,gk1K∑k=1K∑i=1I∑g=1GJi,gk(18b)s.t.pi,gk≤pmax,i∈I,g∈G,(18c)Ti,gtotal,k≤Ti,gmax,i∈I,g∈G,(18d)⋃i∈I⋃g∈GUig=N,i∈I,g∈G,(18e)Ug∩Ug′=⌀,g,g′∈G,


Constraint (Equation 18b) specifies that the transmission power of sensor *i* in channel *g* must be less than the maximum transmission power limits. Constraint (18c) ensures that the overall transmission time experienced by sensor *i* over channel *g* does not exceed the prescribed maximum threshold. Constraints (Equation 18d) and (Equation 18e) indicate that each sensor can be assigned to only one channel.

## 3. Solution of the Optimization Probelm

In this chapter, we propose an optimization method based on algorithm decomposition for the MINLP problem in Problem 2. Due to the high computational complexity typically associated with solving MINLP problems, we decompose the original problem into two subproblems to reduce the computational burden during the solution process, aiming to improve both the solving efficiency and practical feasibility of the implementation.

### 3.1. Grouping Strategy Based on Coalition Game Theory

To obtain the initial grouping $U_{ini}$, sensors are uniformly and randomly assigned to each channel, achieving an even distribution of sensors across channels (see Algorithm 1).

In the above system model, the sensors are divided into multiple channels, each forming a coalition of the same size. Within each coalition, sensors collaboratively offload their respective data blocks to the base station, while competition exists between different coalitions. The objective of the optimization problem is to maximize the overall system performance. We define the outage risk under the joint optimization of remote state estimation and transmission delay as the utility function *v* within the coalition, representing the total outage risks of all sensors within the coalition, as follows:(19)$$v\left(G\right)=\left(O_{1},O_{2},\ldots ,O_{N}\right).$$

Before the matching process, the mutual preference lists are indispensable. Both sensors and base station need to build their preference lists based on their own utilities. We define Γ as the total outage risks of all sensors within a coalition. For any sensors $U_{i}^{g}\in N$, $U_{j}^{g^{\prime }}\in N$ with $U_{i}^{g}\in G_{g}$ and $U_{j}^{g^{\prime }}\in G_{g^{\prime }}$, $G_{g}{≻}_{U_{i}^{g},U_{j}^{g^{\prime }}}G_{g^{\prime }}$ indicates that sensor $U_{i}^{g}$ prefers joining coalition $G_{g}$ over $G_{g^{\prime }}$. In this case, the sensor will exchange positions with those in $G_{g^{\prime }}$. The final result is that after exchanging positions, the total outage risks of sensors in different coalitions is higher than before the exchange. The new partition can be denoted as $\Gamma \left\{G^{\prime }\right\}=\Gamma \left(G_{g}\backslash \left\{U_{i}^{g}\right\}\cup \left\{U_{j}^{g^{\prime }}\right\}\right)+\Gamma \left(G_{g^{\prime }}\backslash \left\{U_{j}^{g^{\prime }}\right\}\cup \left\{U_{i}^{g}\right\}\right)$.
**Algorithm 1** Intelligent Sensor Grouping Algorithm Based on Coalition Game Theory under Fixed Power Constraints**Input:** N, $G^{0}$, I, $h_{i}^{g}$, $p_{i}^{g}$, n∈N, i∈I, $g\in G^{0}$.**Output:** Optimal grouping $G_{fin}$.**Initialization:** t=1, $p_{i}^{g}=p_{\max}$, $U_{ini}=U_{i}^{g}$, $G^{0}=G_{ini}$, ∀i∈I, $g\in G^{0}$.1:**while** $G^{t}$ does not converge to Nash-stable partition $G_{fin}$ **do**2:   Select $U_{i}^{g}\in G_{g}$ and $U_{j}^{g^{\prime }}\in G_{g^{\prime }}$, where $g\neq g^{\prime }$, i∈I, j∈I3:   Define $G^{\prime }=\left(G^{t}\backslash \left\{G_{g^{\prime }},G_{g}\right\}\right)\cup \left\{G_{g^{\prime }}\backslash \left\{U_{i}^{g}\right\}\cup \left\{U_{j}^{g^{\prime }}\right\},G_{g}\backslash \left\{U_{j}^{g^{\prime }}\right\}\cup \left\{U_{i}^{g^{\prime }}\right\}\right\}$4:   Calculate $E\left(G_{m}\right),E\left(G_{m^{\prime }}\right)$ by Problem 45:   **if** $\Gamma \left\{G^{\prime }\right\}>\Gamma \left\{G^{t}\right\}$ **then**6:     Update $G^{t+1}=G^{\prime }$7:     Update t=t+18:   **end if**9:**end while**

### 3.2. Power Allocation Based on Dinkelbach, SCA, and Dual Decomposition Methods

This section addresses the power allocation problem for a fixed sensor grouping by applying the Dinkelbach algorithm, along with the SCA and dual decomposition methods discussed below. To address the issue of offloading time consumption in the expectation, we use the Dinkelbach algorithm to handle the fractional programming. This method is particularly suitable because our objective function involves a ratio structure, and Dinkelbach’s algorithm efficiently handles such problems with convergence guarantees. The transformed problem is described as follows (see Algorithm 2):(20)$$J_{i,g}^{k}=\mathrm{Tr}\left\{Z_{i,g}^{k}-\prod_{z=1}^{i}e^{-\frac{\phi_{z,g}^{k}}{2\mu^{2}}}\left(\lambda_{i,g}^{k}\frac{R_{i,g}^{k}}{\xi }+L_{i,g}^{k}\right)\right\}.$$
where $Z_{i,g}^{k}$ and $L_{i,g}^{k}$ represent $\xi T_{i,g}^{local,k}+\left(1-\xi \right)\psi {\bar{P}}_{i}^{g}$ and $\left(1-\xi \right)\psi Z_{i,g}^{k}-D_{i}^{g}-\xi T_{i,g}^{base,k}$, respectively, and $\lambda_{i,g}^{k}$ is a non-negative parameter. We define $F\left(\lambda_{i,g}^{k}\right)=-\min_{p_{i,g}^{k}}\left\{\lambda_{i,g}^{k}\frac{R_{i,g}^{k}}{\xi }-D_{i}^{g}\right\}$.
**Algorithm 2** Iterative Algorithm Based on Dinkelbach**Input:** $\xi ,D_{i}^{g},R_{i,g}^{k},\text{iter\_max},\epsilon ,i\in I,g\in G$.**Initialization:** $\lambda_{i,g}^{\mathrm{iter},k}=0,\mathrm{iter}=1$.1:**while** 
$F(\lambda_{i,g}^{\mathrm{iter},k})>\epsilon$ 
**or** 
iter≤iter_max 
**do**2:   $F\left(\lambda_{i,g}^{\mathrm{iter},k}\right)=D_{i}^{g}-\lambda_{i,g}^{\mathrm{iter},k}\frac{R_{i,g}^{k}}{\xi }$3:   iter=iter+14:   $\lambda_{i,g}^{\mathrm{iter},k}=\xi \frac{D_{i}^{g}}{R_{i,g}^{k}}$5:**end while****Output:** Obtain $\lambda_{i,g}^{k}=\lambda_{i,g}^{\mathrm{iter},k}$.

To facilitate subsequent analysis, we can derive from Equation (Equation 20) that the minimization problem above can be transformed into a maximization problem. Additionally, since $Z_{i}^{g}\left(k\right)$ is a constant, it has no impact on sensor grouping or power allocation. We redefine an objective function $O_{i,g}^{k}$, which is derived from the original objective function $J_{i,g}^{k}$ by excluding the $Z_{i,g}^{k}$ component and can be considered an approximation of $J_{i,g}^{k}$.(21)$$O_{i,g}^{k}\approx -J_{i,g}^{k}=\mathrm{Tr}\left\{\prod_{z=1}^{i}e^{-\frac{\phi_{z,g}^{k}}{2\mu^{2}}}\left(\lambda_{i,g}^{k}\frac{R_{i,g}^{k}}{\xi }+L_{i,g}^{k}\right)\right\}.$$

Therefore, Problem 1 can be reformulated as(22)$$\begin{array}{c} \begin{array}{cc} \mathrm{Problem}\;2: & \max_{p_{i,g}^{k}}\frac{1}{K}\sum_{k=1}^{K}\sum_{i=1}^{I}\sum_{g=1}^{G}O_{i,g}^{k}\\ s.t. & (18b),(18c). \end{array} \end{array}$$

Moreover, since the subproblem of outage probability involves a product of terms, it is challenging to solve directly. To address this, we apply a natural logarithm function to transform the product into a summation, thereby simplifying the problem and facilitating subsequent analysis.(23)$$\begin{array}{cc} E_{i,g}^{k} & =\ln O_{i,g}^{k}=\mathrm{Tr}\left\{\ln\left(\frac{\lambda_{i,g}^{k}R_{i,g}^{k}}{\xi }+L\right)\right.\\ \; & \left.+\sum_{z=1}^{i}\frac{\sum_{j=z+1}^{I}p_{j,g}^{k}|h_{j,g}^{k}{)|}^{2}+\sigma^{2}}{-p_{z,g}^{k}}F_{z,g}^{k}\right\}, \end{array}$$
where $F_{z,g}^{k}=\frac{\left(2^{\frac{{\hat{R}}_{z}^{g}}{B}}-1\right)\left({\left(1+d_{z,g}^{k}\right)}^{\beta }\right)}{2\mu^{2}}.$

Problem 2 is rewritten as(24)$$\begin{array}{c} \begin{array}{cc} \mathrm{Problem}\;3: & \max_{p_{i,g}^{k}}\frac{1}{K}\sum_{k=1}^{K}\sum_{i=1}^{I}\sum_{g=1}^{G}E_{i,g}^{k}\\ s.t. & (18b),(18c). \end{array} \end{array}$$

Since Problem 3 is a non-convex optimization problem with respect to $p_{i,g}^{k}$, it is challenging to solve directly. Therefore, we adopt the SCA method, whose convergence is proven in [11]. Hence, a lower bound for $R_{i,g}^{k}$ can be expressed as(25)$$\begin{array}{cc} R_{i,g}^{k} & =B\log_{2}\left(1+{\mathrm{SIN}R}_{i,g}^{k}\right)\\ \; & \geq B\left(a_{i,g}^{k}\log_{2}\left({\mathrm{SIN}R}_{i,g}^{k}\right)+b_{i,g}^{k}\right)={\hat{R}}_{i,g}^{k}, \end{array}$$
where the equality holds when(26)$$a_{i,g}^{k}=\frac{{\mathrm{SIN}R}_{i,g}^{k}}{1+{\mathrm{SIN}R}_{i,g}^{k}},$$
and(27)$$b_{i,g}^{k}=\log_{2}\left(1+{\mathrm{SIN}R}_{i,g}^{k}\right)-a_{i,g}^{k}\log_{2}\left({\mathrm{SIN}R}_{i,g}^{k}\right).$$

With the fixed approximation coefficients $A\left(k\right)≜a_{i}^{g}\left(k\right)$ and $B\left(k\right)≜b_{i,g}^{k}$, the constraint in Problem 3 remains non-concave with respect to $p_{i,g}^{k}$. Therefore, we introduce the transformation ${\bar{p}}_{i,g}^{k}=\ln p_{i,g}^{k}$.

Here, we decompose the objective function in Problem 3 into two parts, $U_{i,g}^{k}$ and $D_{i,g}^{k}$, which are expressed as follows:(28)$$E_{i,g}^{k}=\mathrm{Tr}\left\{U_{i,g}^{k}+D_{i,g}^{k}\right\},$$(29)$$U_{i,g}^{k}=\sum_{z=1}^{i}\frac{\sum_{j=z+1}^{I}e^{{\bar{p}}_{j,g}^{k}}{\left|h_{j,g}^{k}\right|}^{2}+\sigma^{2}}{-e^{{\bar{p}}_{z,g}^{k}}}F_{z,g}^{k},$$
and(30)$$\begin{array}{cc} D_{i,g}^{k} & =\ln\left(\frac{\lambda_{i,g}^{k}R_{i,g}^{k}}{\xi }+L_{i,g}^{k}\right)\\ \; & =\ln\left\{\frac{\lambda_{i,g}^{k}a_{i,g}^{k}{\bar{p}}_{i,g}^{k}{\left|h_{i,g}^{k}\right|}^{2}}{\xi \ln2}+S_{i,g}^{k}\right\}, \end{array}$$
where $\frac{\lambda_{i,g}^{k}a_{i,g}^{k}\ln\left(\sum_{j=i+1}^{I}e^{{\bar{p}}_{j,g}^{k}}{\left|h_{j,g}^{k}\right|}^{2}+\sigma^{2}\right)+b_{i,g}^{k}}{-\xi \ln2}+L_{i,g}^{k}$ is abbreviated as $S_{i,g}^{k}$.

The first-order partial derivative of $U_{i,g}^{k}$ is given as follows:(31)$$\frac{\partial U_{i,g}^{k}}{\partial ({\bar{p}}_{i,g}^{k})}=\left\{\begin{array}{c} \frac{|h_{i,g}^{k}{|}^{2}}{V_{i,g}^{k}}F_{i,g}^{k},\\ -\sum_{z=1}^{i-1}\delta_{z,g}^{k}F_{z,g}^{k}+\frac{|h_{i,g}^{k}{|}^{2}}{V_{i,g}^{k}}F_{i,g}^{k},\\ -\sum_{z=1}^{i-1}\delta_{z,g}^{k}F_{z,g}^{k}+\frac{\sigma^{2}}{e^{{\bar{p}}_{i,g}^{k}}}F_{i,g}^{k}, \end{array}\right.$$
where we define $V_{i,g}^{k}=\frac{e^{{\bar{p}}_{i,g}^{k}}{\left|h_{i,g}^{k}\right|}^{2}}{\sum_{j=i+1}^{I}e^{{\bar{p}}_{j,g}^{k}}{\left|h_{j,g}^{k}\right|}^{2}+\sigma^{2}}$ and $\delta_{z,g}^{k}=\frac{e^{{\bar{p}}_{i,g}^{k}}{\left|h_{i,g}^{k}\right|}^{2}}{e^{{\bar{p}}_{z,g}^{k}}}$. The three cases, listed from top to bottom, correspond to i=1, 1<i<I, and i=I, respectively.

The second-order partial derivative of $U_{i}^{g}\left(k\right)$ is given by(32)$$\frac{\partial^{2}U_{i,g}^{k}}{\partial {\left({\bar{p}}_{i,g}^{k}\right)}^{2}}=\left\{\begin{array}{c} -\frac{|h_{i,g}^{k}{|}^{2}}{V_{i,g}^{k}}F_{i,g}^{k},\\ -\sum_{z=1}^{i-1}\delta_{z,g}^{k}F_{z,g}^{k}-\frac{|h_{i,g}^{k}{|}^{2}}{V_{i,g}^{k}}F_{i,g}^{k},\\ -\sum_{z=1}^{i-1}\delta_{z,g}^{k}F_{z,g}^{k}-\frac{\sigma^{2}}{e^{{\bar{p}}_{i,g}^{k}}}F_{i,g}^{k}, \end{array}\right.$$
where the three constraint conditions for the second-order partial derivatives are the same as those for the first-order derivatives.

The first- and second-order derivatives of $D_{i}^{g}\left(k\right)$ are expressed as follows(33)$$\frac{\partial D_{i,g}^{k}}{\partial {\bar{p}}_{i,g}^{k}}=\frac{\frac{\lambda_{i,g}^{k}a_{i,g}^{k}{\left|h_{i,g}^{k}\right|}^{2}}{\xi \ln2}}{\frac{\lambda_{i,g}^{k}a_{i,g}^{k}{\left|h_{i,g}^{k}\right|}^{2}}{\xi \ln2}{\bar{p}}_{i,g}^{k}+S_{i,g}^{k}},$$(34)$$\frac{\partial^{2}D_{i,g}^{k}}{\partial {\left({\bar{p}}_{i,g}^{k}\right)}^{2}}=\frac{-{\left(\frac{\lambda_{i,g}^{k}a_{i,g}^{k}{\left|h_{i,g}^{k}\right|}^{2}}{\xi \ln2}\right)}^{2}}{{\left(\frac{\lambda_{i,g}^{k}a_{i,g}^{k}{\left|h_{i,g}^{k}\right|}^{2}}{\xi \ln2}{\bar{p}}_{i,g}^{k}+S_{i,g}^{k}\right)}^{2}}.$$

According to Equations (Equation 31)–(Equation 34), Problem 3 is proven to be a convex optimization problem, and we use the dual decomposition method to solve it. Consequently, the Lagrangian function for Problem 4 is(35)$$\begin{array}{cc} \; & \sum_{i=1}^{I}\sum_{g=1}^{G}L\left(p_{i,g}^{k},\eta_{i,g}^{k},\phi_{i,g}^{k}\right)\\ = & \sum_{i=1}^{I}\sum_{g=1}^{G}E_{i,g}^{k}+\sum_{i=1}^{I}\sum_{g=1}^{G}\eta_{i,g}^{k}\left(p_{\max}-p_{i,g}^{k}\right)\\ + & \sum_{i=1}^{I}\sum_{g=1}^{G}\phi_{i,g}^{k}\left(T_{i,g}^{max}-T_{i,g}^{total,k}\right), \end{array}$$
where $\eta_{i,g}^{k}$ and $\phi_{i,g}^{k}$ are the Lagrangian multipliers.

According to (Equation 35), the dual function $D\left(\eta_{i,g}^{k},\phi_{i,g}^{k}\right)$ can be expressed as(36)$$\begin{array}{ccc} D\left(\eta_{i,g}^{k},\phi_{i,g}^{k}\right) & = & \left\{\begin{array}{c} \max_{{\bar{p}}_{i,g}^{k}}L\left({\bar{p}}_{i,g}^{k},\eta_{i,g}^{k},\phi_{i,g}^{k}\right)\\ \;s.t.\;{\bar{p}}_{i,g}^{k}\geq 0. \end{array}\right. \end{array}$$

Hence, the dual problem is(37)$$\begin{array}{c} \begin{array}{cc} \mathrm{Problem}\;4: & \min D\left(\eta_{i,g}^{k},\phi_{i,g}^{k}\right)\\ s.t. & \eta_{i,g}^{k}\geq 0,\phi_{i,g}^{k}\geq 0. \end{array} \end{array}$$

The power allocation ${\bar{p}}_{i,g}^{k}$ for the fixed values $\eta_{i,g}^{k}$, $\phi_{i,g}^{k}$ is calculated via (Equation 37) by applying the KKT condition on (Equation 35). Hence, we have (see Alogorithm 4)(38)$$\frac{\partial L\left(p_{i,g}^{k},\eta_{i,g}^{k},\phi_{i,g}^{k}\right)}{\partial {\bar{p}}_{i,g}^{k}}=0.$$

Since the problem involves solving a multi-objective nonlinear equation system where the result is equal to zero, we utilized the lsqnonlin function in MATLAB 2024b to find the optimal solution ${\bar{p}}_{i,g}^{*,k}$.

For $\eta_{i,g}^{k}$ and $\phi_{i,g}^{k}$, we use the gradient descent method to obtain the optimal values:(39)$$\eta_{i,g}^{j+1,k}={\left[\eta_{i,g}^{j,k}-\Delta \varepsilon_{1}^{j,k}\left(p_{\max}-p_{i,g}^{k}\right)\right]}^{+},$$(40)$$\phi_{i,g}^{j+1,k}={\left[\phi_{i,g}^{j,k}-\Delta \varepsilon_{2}^{j,k}\left(T_{i,g}^{max,k}-T_{i,g}^{total,k}\right)\right]}^{+}.$$
where *j* is an iteration index. The parameters $\Delta \varepsilon_{1}$ and $\Delta \varepsilon_{2}$ represent the learning rates for $\eta_{i,g}^{j}$ and $\phi_{i,g}^{j}$, respectively.

### 3.3. Complexity Analysis

The computational complexity of the proposed algorithms was analyzed as follows: The complexity is determined by the number of iterations of the while loop, combined with the nested loops, resulting in a time complexity of $O(t_{\max}\cdot K\cdot I\cdot G)$, where $t_{\max}$ and *K* represent the maximum number of iterations and the iteration count within the loop, respectively; *G* is the number of groups; and *I* is the number of sensors within each group. For Algorithm 3, the complexity is approximately $O(T\cdot N^{2}\cdot G)$, where *T* denotes the number of iterations required to achieve a Nash-stable partition and *N* represents the total number of sensors. For Algorithms 3, the complexity is O(K·L·M), where *L* is the number of iterations required for convergence and *M* is the number of power allocation variables. Thus, the total complexity of the system, combining the complexities of sensor grouping and power allocation processes, is $O(t_{\max}\cdot K\cdot I\cdot G+T\cdot N^{2}\cdot G+K\cdot L\cdot M).$ To further verify the practical efficiency of the proposed algorithm, we report the actual computational time of each method under various system scales, as summarized in Table 2.
**Algorithm 3** Power Allocation Based on SCA and Dual Decomposition Methods**Input:** $\lambda_{i,g}^{\mathrm{iter},k},\xi ,\psi ,\text{iter\_max},\epsilon ,D_{i}^{g},R_{i,g}^{k},\sigma ,i\in I,g\in G$.**Initialization:** $a_{i,g}^{k}$ = $b_{i,g}^{k}$ = 1, Δε = $\Delta \varepsilon_{2}$ = 0.1, $p_{i,g}^{k}$ = $p_{\max}$, *j* = iter = 1, $E_{i}^{g}$ = $E_{i,g}^{k}=0$.1:**for** k=1 to *K* **do**2:   **while** Δ>ϵ **or** iter≤iter_max **do**3:     Update iter=iter+14:     Update $a_{i,g}^{k}$ according formula (Equation 26)5:     Update $b_{i,g}^{k}$ according formula (Equation 27)6:     Calculate $p_{i,g}^{*,k}=\exp\left({\bar{p}}_{i,g}^{*,k}\right)$7:     **repeat**8:        j=j+19:        Calculate ${\bar{p}}_{i,g}^{k}$ according to formula (Equation 38)10:        Update $\left\{\eta_{i,g}^{k}\right\}$ according to formula (Equation 39)11:        Update $\left\{\phi_{i,g}^{k}\right\}$ according to formula (Equation 40)12:     **until** j≥iter_max13:     Update $\Delta =|F\left(\lambda_{i,g}^{\mathrm{iter},k}\right)-F\left(\lambda_{i,g}^{\mathrm{iter}-1,k}\right)|$14:   **end while**15:   Obtain $E_{i,g}^{k}$ according to formula (Equation 23)16:   Update $E_{i}^{g}=E_{i}^{g}+E_{i,g}^{k}$17:**end for****Output:** $E_{i}^{g}=E_{i,g}^{k}/K$.

## 4. Performance Evalution

In this section, we evaluate the performance of the resource allocation algorithm for minimizing outage risk under the joint optimization of remote state estimation and transmission delay. The experiments were conducted using MATLAB 2024b. The simulation setup positioned the base station at coordinates (0m,0m), with sensors randomly distributed within a 100m×100m area. This area size was chosen to represent a typical industrial Internet of Things (IIoT) scenario, where a large number of sensors operate in a compact space with short-range wireless communication and strict latency and reliability requirements. We simulated 1000 Rayleigh fading channel realizations and used the sampled values to calculate channel gains. Each sensor was assigned a bandwidth of 1KHz for data transmission. To reflect a dynamic and time-varying wireless environment, we further assumed that sensor positions were updated at each time slot, simulating node mobility or deployment variations commonly encountered in practical wireless sensor networks. Specific parameters are summarized in Table 3. The system parameters $A_{i}^{g}$, $C_{i}^{g}$, $Q_{i}^{g}$, and $R_{i}^{g}$ were sampled from the range [0.5,1.5] to reflect moderate system dynamics and noise levels while ensuring numerical stability for Kalman filtering. The optimization-related parameters were set as ζ=0.4, ψ=1, and μ=1, representing a balanced trade-off between communication delay and estimation accuracy, as well as a standard Rayleigh fading environment. These parameter settings were consistent with prior works and supported the generality and reliability of the simulation.

For the sensor grouping algorithm, we evaluated two alternative strategies and compared them with the proposed heuristic algorithm, which optimized sensor grouping under fixed power constraints to minimize the state estimation outage risk. The two strategies were random sensor grouping [28] and a suboptimal grouping approach [29]. The suboptimal strategy operated as follows: given *N* sensors, they were first sorted in descending order based on their channel gains. The sorted sensors were then evenly divided into *M* blocks. Sensors from the same position in each block formed a group (e.g., the first sensor in each block formed the first group, the second sensor formed the second group, etc.) until all groups were created. For the power allocation algorithm, we compared two approaches: random power allocation within the allowable range [30] and assigning the maximum power value to all sensors [15]. Comparing the algorithm with the four aforementioned algorithms helped evaluate the performance, computational complexity, and practical feasibility of different strategies in the uplink NOMA system, thereby optimizing resource allocation.

Figure 2 presents the sum of the expected function in Problem 4 with nine sensors and three channels. The simulation parameters were set as ξ=0.5, $\hat{R}=100\;\mathrm{bps}$, B=1KHZ, and $p_{\max}=20\;\mathrm{dBm}$. We compared different sensor grouping strategies, including the proposed coalition game-based intelligent grouping algorithm, the random grouping algorithm, and the suboptimal grouping algorithm. As shown, across all time slots *k*, the coalition game-based algorithm consistently outperformed the other two in solving the sum of expectations in Problem 1. Each point was obtained from 500 Monte Carlo simulations per algorithm per time slot.

Figure 3 compares different power allocation methods under the same coalition game-based grouping strategy. Simulation settings and Monte Carlo parameters were consistent with those in Figure 2. The joint Dinkelbach algorithm, the SCA algorithm, and the KKT-based power allocation were evaluated against random and full power allocation. The proposed joint allocation methods clearly outperformed the baselines in solving the sum of expectations in Problem 1.

Figure 4 illustrates the impact of different minimum transmission rates on the total value of the objective function in Problem 1. The minimum rates were set to 100bps, 500bps, and 1000bps, while other parameters remained unchanged. It was observed that the total objective value increased significantly with ${\hat{R}}_{i,g}^{k}$, which aligned with the theoretical expectations based on Equation (Equation 20).

Figure 5 examines the effect of bandwidth *B* on $J_{i,g}^{k}$. Bandwidth values were set to 1KHz, 10KHz, and 100KHz. The curves show that as *B* increased, $J_{i,g}^{k}$ grew accordingly. Limited bandwidth constrained communication performance, while a larger *B* improved transmission rates and system efficiency, validating the model’s sensitivity to bandwidth variations.

In Figure 6, we show the impact of the weighting parameter ξ on the total value $J_{i,g}^{k}$ of the objective function in Problem 1. Specifically, we varied the parameter ξ with values ξ=0.2, ξ=0.5, and ξ=0.8. The results indicate a clear trend: as the proportion allocated to ξ increased, the total value $J_{i,g}^{k}$ also rose. This suggests that in the joint optimization of remote state estimation using KF and the offloading time consumption, the latter has a more pronounced influence on the objective function. Given that our optimization problem aimed to minimize the objective function, this result highlights a critical trade-off. To achieve optimal system performance, it is advisable to assign a relatively smaller weight to offloading time consumption while allocating a greater proportion to the remote state estimation component. This allocation strategy aligns with the objective of minimizing the overall efficiency and benefit of the system.

In Figure 6, the impact of the weighting parameter ξ on $J_{i,g}^{k}$ is shown. We considered 0.2, 0.5, and 0.8. The results show that as ξ increased, the total objective value rose. This indicates that offloading time consumption has a greater influence on the objective, suggesting that assigning a smaller weight to offloading time can better optimize system performance.

Figure 7 analyzes the impact of maximum transmission power on $J_{i,g}^{k}$. As $p_{\max}$ increased, the total objective value decreased. This was because higher transmission power strengthened the transmitted signal, thereby improving transmission reliability according to the Shannon–Hartley theorem. However, the resulting increase in power consumption must be considered in practical applications to balance performance and energy efficiency.

In Figure 8, we analyze how different sensor allocations within each channel group affected $J_{i,g}^{k}$ with a fixed total of 12 sensors. Group sizes of two, three, and four sensors were compared. The results show that increasing the number of sensors per group improved system performance by reducing the total objective value. However, it significantly increased computational complexity, with computation time for four sensors per group being approximately 10 times higher than for two or three sensors. Thus, a trade-off between performance and computational cost must be carefully considered.

## 5. Conclusions

This paper investigated the problem of minimizing outage risk under joint optimization of remote state estimation and transmission delay. By formulating the problem as a MINLP problem, it was further divided into an intelligent sensor grouping algorithm based on coalition game theory and a joint power allocation algorithm leveraging the Dinkelbach algorithm, SCA, and dual decomposition methods. Simulation results demonstrate that the proposed algorithm outperforms traditional sensor grouping and power allocation algorithms in terms of performance, effectively reducing the outage risk of sensors during transmission. Future research can further extend to multi-sensor and multi-base station scenarios in downlink NOMA systems, exploring outage risk assessment methods in more complex systems.

## Figures and Tables

**Figure 1 sensors-25-04686-f001:**
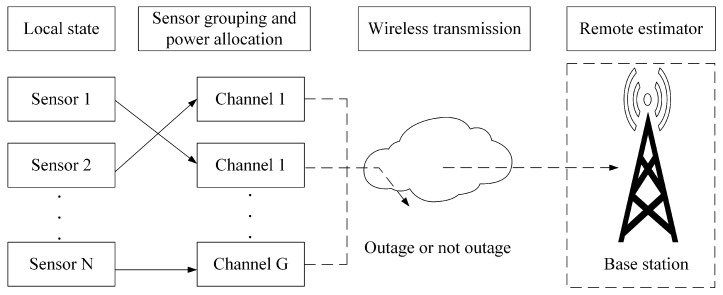
System model of remote state estimation outage risk.

**Figure 2 sensors-25-04686-f002:**
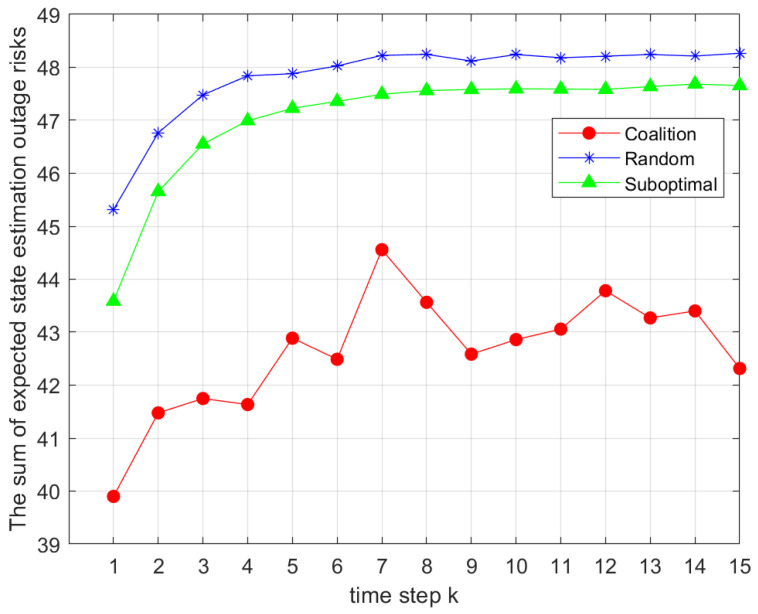
Comparison of the total outage risks under different sensor grouping strategies.

**Figure 3 sensors-25-04686-f003:**
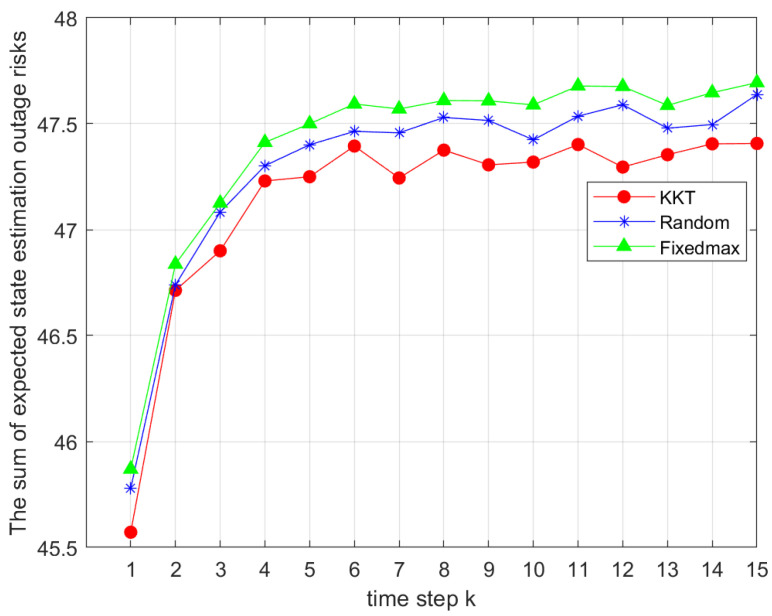
Comparison of the total outage risks under different power allocation strategies.

**Figure 4 sensors-25-04686-f004:**
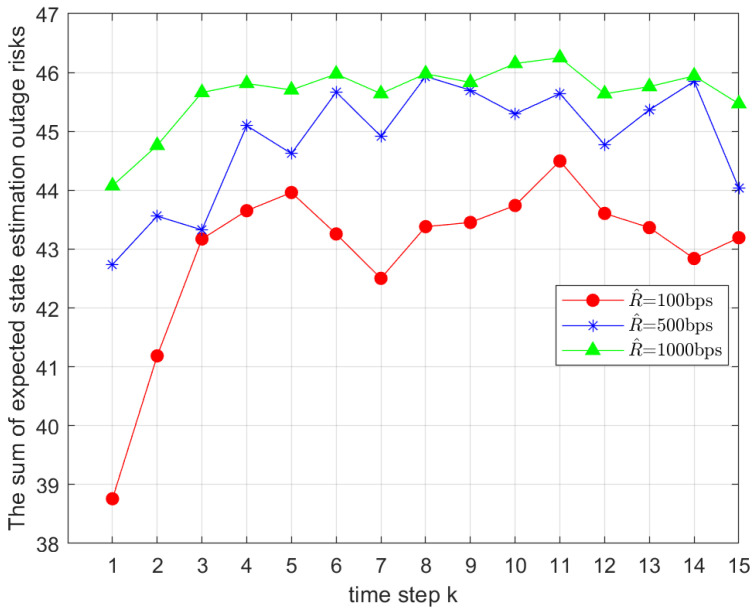
Comparison of the total outage risks under different minimum transmission rates $\hat{R}$.

**Figure 5 sensors-25-04686-f005:**
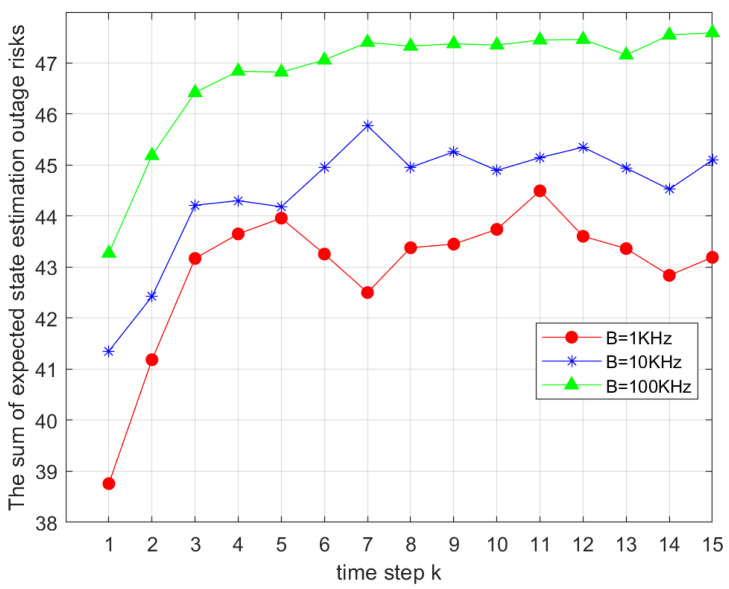
Comparison of the total outage risks under different bandwidths *B*.

**Figure 6 sensors-25-04686-f006:**
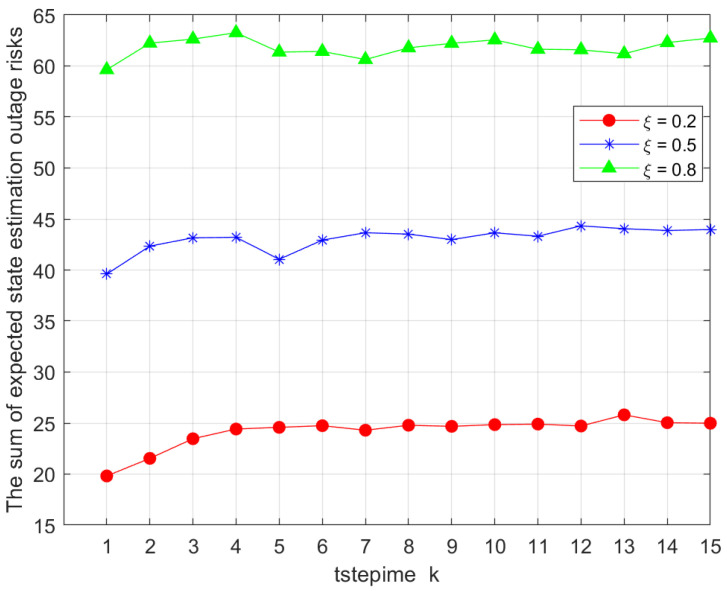
Comparison of total outage risks under different association factors ξ.

**Figure 7 sensors-25-04686-f007:**
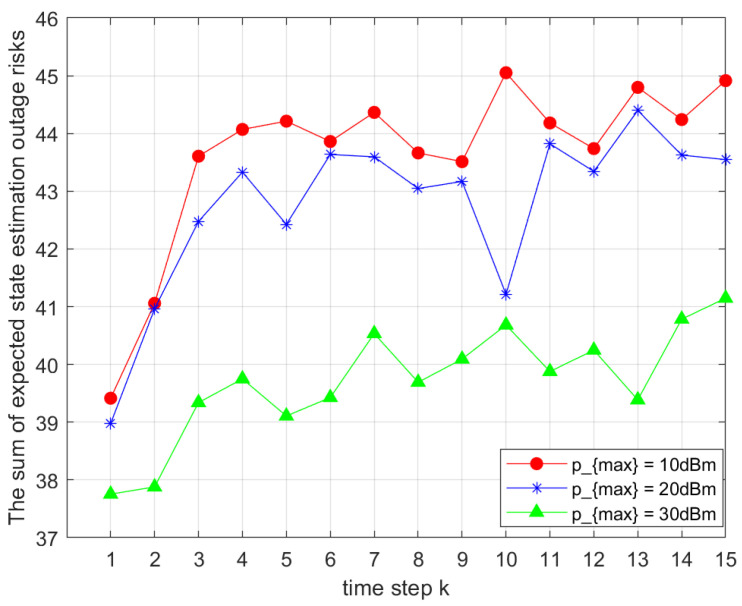
Comparison of total outage risks under different maximum powers $p_{\max}$.

**Figure 8 sensors-25-04686-f008:**
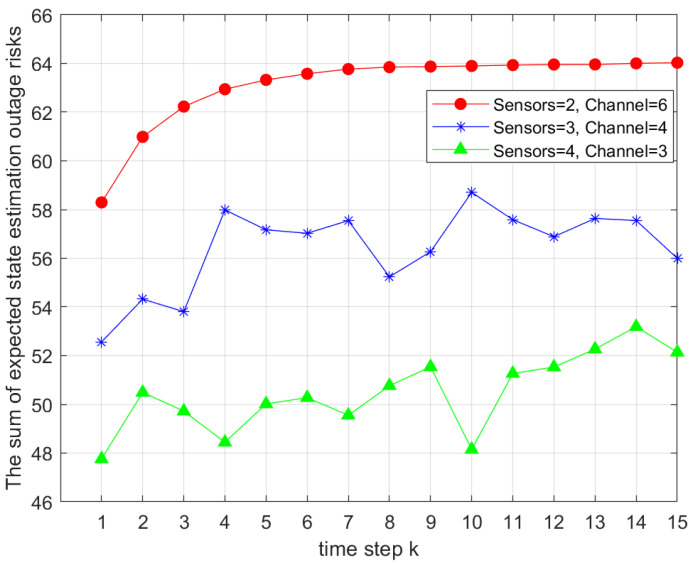
Comparison of total outage risks under different sensor counts within the same channel.

**Table 1 sensors-25-04686-t001:** Comparison of the proposed method with existing approaches.

Feature	In [26]	In [22]	Proposed Method
Sensor grouping	Not considered	Heuristic clustering	Game-theoretic
Power allocation	Fixed allocation	Cognitive + Fairness	Dinkelbach + SCA
Outage	Not addressed	Closed-form	Closed-form
Estimation	KF	Ignored	KF

**Table 2 sensors-25-04686-t002:** Performance comparison under different system scales.

System Scale (N,I,G)	This Work	Random	Suboptimal
(12, 2, 6)	1846.6s	1739.0s	1793.1s
(12, 3, 4)	1622.3s	1565.7s	1583.0s
(12, 4, 3)	1571.0s	1494.9s	1525.4s

**Table 3 sensors-25-04686-t003:** Summary of the simluation parameters.

Parameters	Values
Simulation area size	100 m × 100 m
Base station antenna location	(0, 0) m
CPU cycles required to process task $I_{i}^{g}$	1000 Megacycles
Computational capability of sensor $F^{s}$	1 GHz
Computational capability of base station $F^{b}$	10 GHz
System bandwith *B*	1 KHz
Minimum transmission rate $\hat{R}$	100 bps
Maximum power $p_{\max}$	20 dBm
Maximum tolerable transmission delay $T_{i,g}^{max}$	0.1 s
Size of received input data $D_{i}^{g}$	80 B
Noise power density σ	−170 dBm/Hz
State transition matrix $A_{i}^{g}$	[0.5, 1.5]
Measurement matrix $C_{i}^{g}$	[0.5, 1.5]
Process noise covariance $Q_{i}^{g}$	[0.5, 1.5]
Measurement noise covariance $R_{i}^{g}$	[0.5, 1.5]
Weighting factor ξ	0.4
Scaling factor ψ	1
Variance of the normal distribution μ	1

## Data Availability

The data that support the findings of this study are available from the corresponding author upon reasonable request.

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
