# Peer review of "Joint Optimization of Radio and Computational Resource Allocation in Uplink NOMA-Based Remote State Estimation"

_sensors, 2025, doi:10.3390/s25154686_

Round 1
Reviewer 1 Report
Comments and Suggestions for Authors
1. In the related work section, add a table comparing the contribution of the proposed approach with other existing approaches.
2. In equation 22, a more detailed explanation of the parameters ξ and 𝜓 is needed. Furthermore, if possible, include experiments on the sensitivity to the parameters ξ and 𝜓. In this paper, the values 𝜉=0.4 or 𝜓=1 are used. Why were these values chosen? Are they optimal?
3. In 5G and 6G networks, simulations should be conducted for moving nodes. However, in this article, simulations are only conducted in static scenarios. If possible, add experiments with dynamic scenarios to demonstrate the robustness of the proposed algorithm to changing environmental conditions.
4. In the complexity analysis section, add graphs showing the computational time required for each method.
5. Some references are too old; add some recent references from reputable journals.
Reviewer 2 Report
Comments and Suggestions for Authors
1) The set of sensor indices in each cluster is denoted by "I", but it appears that this set may vary across clusters. The notation should be clarified or revised for consistency.
2) The impulse response of the channel between sensor i and the base station, h_i,g^k, is modeled as:
h_i,g^k = e_i,g^k / sqrt(1 + (d_i,g^k)^beta)
The reviewer is curious about the rationale for using the square root and the specific form "1 + (d_i,g^k)^beta" in this model.
3) How is Equation (15) derived from Equation (14), particularly when gamma_i,g^k = 0? Please provide a detailed explanation.
4) The objective in Equation (22) is described as jointly optimizing sensor grouping and power allocation. Could the authors clarify the exact meaning of this objective? Does it represent the total transmission power?
5) The original optimization problem in Equation (22) is decomposed into two subproblems to reduce computational complexity. How can we ensure that the resulting solution remains suboptimal or near-optimal?
6) Following the same reasoning as in comment (5), the Dinkelbach algorithm is used to solve the fractional programming problem. Are there alternative methods to handle such problems? Please justify the choice of the Dinkelbach algorithm.
7) In the evaluation section, the reviewer would like to understand the rationale behind the selected values for the optimization problem parameters. Specifically, why are A_i^g, C_i^g, Q_i^g, and R_i^g chosen within the range [0, 5, -1.5]? Also, why are the parameters zeta = 0.4, psi = 1, and mu = 1?
8) Additionally, the simulation area is set to 100m x 100m, which seems small for a typical cellular coverage area. What type of cellular scenario is being targeted in this study?
Round 2
Reviewer 2 Report
Comments and Suggestions for Authors
The authors have been well addressed my comments.